# Exercise prescription for the prevention and treatment of chronic diseases in primary care: Protocol of the RedExAP study

Alicia Saz-Lara[1], José Alberto Martínez Hortelano[1,2,3]*, María Medrano[4,5], Raquel Luengo-González[2,6], Miriam Garrido Miguel[1,7], Montserrat García-Sastre[2], José Ignacio Recio-Rodriguez[8], Daniel Lozano-Cuesta[2], Iván Cavero-Redondo[1,9]

1 Health and Social Research Center, Universidad de Castilla-La Mancha, Cuenca, Spain, 2 Nursing and Physiotherapy Department, University of Alcalá, Alcalá de Henares, Spain, 3 Group for Research in Community Care and Social Determinants of Health, Madrid, Spain, 4 Department of Health Sciences, Institute for Innovation & Sustainable Food Chain Development, Public University of Navarra, Pamplona, Spain, 5 CIBER de Fisiopatología de la Obesidad y Nutrición (CIBEROBN), Instituto de Salud Carlos III, Madrid, Spain, 6 Group for Research in Nursing Care, Gregorio Marañón, Health Research Institute (IiSGM), Madrid, Spain, 7 Universidad de Castilla-La Mancha, Faculty of Nursing, Albacete, Spain, 8 Facultad de Enfermería y Fisioterapia (Universidad de Salamanca), Unidad de Investigación en Atención Primaria de Salamanca (APISAL), Instituto de Investigación Biomédica de Salamanca (IBSAL), Red de Investigación en Cronicidad, Atención Primaria y Promoción de la Salud (RICAPPS), Salamanca, Spain, 9 Facultad de Ciencias de la Salud, Universidad Autónoma de Chile, Talca, Chile

* josealberto.martinez@uah.es

**Data Availability Statement:** Deidentified research data will be made publicly available when the study is completed and published.

## Abstract

### Background

Existing evidence supports the effectiveness of exercise in preventing and treating chronic diseases, yet its integration into clinical practice remains limited. This study protocol aims to address the evidence-practice gap by exploring barriers to exercise prescription in primary care and developing a clinical practice guideline (CPG).

### Methods

Employing a qualitative approach, focus groups will be conducted to investigate primary care professionals' challenges in prescribing exercise and patients' adherence to recommendations. Phenomenological analysis will facilitate data interpretation. Data triangulation, expert analysis, and quality criteria will ensure study reliability. The CPG development process is outlined, emphasizing transdisciplinary collaboration and patient involvement.

### Conclusion

The RedExAP study responds to the imperative for evidence-based exercise integration in primary care. The study's combined qualitative exploration and CPG development present the potential to improve health outcomes and cost-effectiveness. By elucidating primary care professionals' and patients' perspectives, the study contributes to enhancing exercise prescription adoption. The innovative transdisciplinary approach aligns with the 2030 Agenda, promoting better population health and greater social well-being, showing promise

**Funding:** This study was funded by the Higher Sports Council through the call "Research Networks in Sports Sciences" for the year 2023, RedExAP network (14/UPB/23) and co-funded by the European Union (ERDF/ESF) and by the Carvascare Research Group fron the Universidad de Castilla-La Mancha (2023-GRIN-34459).

**Competing interests:** The authors have declared that no competing interests exist.

**Abbreviations:** CPG, Clinical Practice Guideline; COREQ, Consolidated Criteria for Reporting Qualitative Research; EtD, Evidence to Decision; PICO, Population, Intervention, Comparator, Outcome; RCT, Randomised Controlled Trial.

in alleviating chronic disease burdens. This study's findings lay the groundwork for advancing evidence-based exercise interventions within primary care to transform chronic disease management.

## Introduction

### Scientific evidence on the influence of physical exercise on health

Since the epidemiologist Jerry Morris, analysing the relationship between physical activity and coronary disease [1], proposed that it was "the best buy" in preventing this pathology, studies in this direction have continued to be published. However, possibly due to unequal research funding, the evidence of exercise benefits in terms of mortality is substantially poorer than pharmacological interventions. Furthermore, surprisingly, trials designed to test the effectiveness of pharmacological interventions rarely compare drugs and physical exercise efficacy [2]. Although it is true that there is a lack of evidence demonstrating the effectiveness of both physical exercise and pharmacological interventions in the prevention of coronary heart disease and diabetes mellitus, it has been shown that physical exercise could be effective in the treatment of patients with stroke, while treatment with diuretics was more effective than exercise in patients with heart failure [2]. Furthermore, there is evidence showing the effectiveness of physical exercise interventions to improve cognition in patients with cognitive impairment and Alzheimer's disease [3].

Therefore, there is a clear asymmetry between the evidence supporting pharmacological treatments and strategies based on physical exercise [3, 4]. Not only are there fewer randomized controlled trials (RCTs) testing the effectiveness of physical exercise, but their sample sizes are substantially smaller, which in turn results in lower statistical power. Thus, there is no robust evidence, and health care providers, despite physical exercise, could potentially be a much more beneficial strategy to opt for pharmacological therapeutic options, even in pathological processes where drugs produce only modest improvements and side effects [2].

### Knowledge transfer to clinical practice

In recent years, numerous published studies have assessed the effectiveness of physical exercise in the preventive or therapeutic management of many prevalent chronic conditions that affect children [5, 6], adults and the elderly [7], and, although the evidence is limited, physical exercise has proven to be superior or potentially similar to the pharmacological treatments commonly used in clinical practice in some cases, such as in patients with stroke, heart failure or diabetes mellitus, among other [2]; however, the reasons why this evidence does not translate into clinical practice remain to be elucidated.

A transdisciplinary approach is needed to include physical activity and sports in health care systems, so the importance of including different specialist healthcare professional profiles (such as cardiologists, geriatricians, rehabilitation medicine physicians, and physiotherapists), primary care physicians and nurses, and physical activity and sports graduates could play a key role in integrating exercise prescription, design, and intervention implementation to prevent and manage chronic diseases. Therefore, although theoretically, health services have incorporated the prescription of physical activity into the range of services they offer, some evidence shows that either this service does not reach users or is provided inadequately [8].

Despite the urgency for this evidenced scientific knowledge to be accessible to healthcare professionals, the efforts made worldwide thus far are scattered and scarce, probably because physical exercise is considered an orphan or unknown drug and, in that case, it should be a priority in public research calls or nonprofit organizations [9].

## Barriers and facilitators for exercise prescription

Although physical exercise prescription is a fundamental pillar in the preventive and therapeutic approach to several clinical entities, primary care professionals show resistance to implementing it [10].

Among the barriers reported by professionals for prescribing physical activity or exercise, the following have been described: lack of appropriate protocols [11], lack of knowledge and self-assurance for prescribing physical exercise [12], deficiencies in counselling skills, beliefs about the low effectiveness and inefficiency of prescribing physical exercise [13], and a tendency to distrust patients' abilities to maintain lifestyle changes [14]. In this sense, many physicians and nurses consider patient motivation for engaging in physical exercise to be important and believe that physical exercise prescription should be carried out by other professionals, such as physiotherapists [15]. Consequently, despite professionals considering it important to increase physical activity levels for better health, physical exercise prescriptions are not being carried out in primary care [16].

However, an increasing number of healthcare professionals show great interest in how to modify patients' habits regarding physical activity [12] and, in general, accept that improving patients' physical condition could improve their health and quality of life [17]. Moreover, many of them report that collaboration between primary care professionals and physical activity professionals can help with physical exercise prescription [18].

Furthermore, patients believe that advice and support in primary care could increase motivation for engaging in physical activity, but the focus is on weight reduction, cardiac issues, and disease-related problems, rather than improving or increasing levels of physical activity per se. They also consider that there is a lack of information regarding physical activity aspects (recommended physical activity levels or the amount of activity that can be safely undertaken according to their health conditions) and that advice from primary care professionals is not sufficiently motivating for older adults and is seen as a short-term strategy [19].

Finally, the economic impact of physical inactivity cost global healthcare systems approximately $67.5 billion in 2013, while in Spain, this figure amounts to approximately €6.9 billion annually, in addition to nearly 20 million working days lost in absenteeism. The costs are mostly borne by public health services, and the predicted trend is for them to increase year after year if effective strategies are not implemented, such as those proposed in the CPG. Behind these numbers are premature deaths and losses in business productivity, but the solution seems clear: investment in physical activity provides a much higher economic return than its cost due to the reduction in public and private spending [20].

Therefore, the RedExAP study aims to i) understand the difficulties that primary care professionals face when prescribing physical exercise to patients with chronic pathologies where physical exercise is an effective prevention or treatment strategy. Additionally, understand the reasons why this therapeutic strategy has been incorporated into the prescription portfolio of those professionals who routinely prescribe physical activity; ii) understand the difficulties that patients in primary care centers perceive in following physical activity and exercise recommendations they receive. Additionally, for those patients who follow the prescribed recommendations, identify the factors that facilitate adherence to recommendations from primary care physicians and nurses; and iii) synthesize, through a clinical practice guideline (CPG) or white

paper, the recommendations on the effectiveness of physical exercise prescription in the prevention and treatment of the most prevalent chronic conditions in Western countries. In this way, when faced with such pathologies, professionals would know if physical exercise is the most effective preventive or therapeutic strategy, the magnitude of the exercise effect, and the most recommended design in terms of type, intensity, frequency, and duration of physical exercise.

## Material and methods

The RedExAP study will be a qualitative study to analyse barriers and facilitators for physical exercise prescription and the results obtained in this study will contribute to the development of a CPG for the effectiveness of physical exercise prescription in the most prevalent chronic conditions.

### Qualitative study

A qualitative study will be conducted to comprehend the challenges faced by primary care professionals (health professionals, such as physician and nurses, among others, who care for people as a whole, in the context of the patient's clinical history and their life circumstances) in prescribing physical exercise to patients and understand the difficulties perceived by patients at primary care centers (health services that satisfy the needs of basic care, and that guarantee comprehensive quality care, from promotion and prevention to treatment, rehabilitation and palliative care) in adhering to received physical activity and exercise recommendations [21]. This study will be designed and analysed through the lens of Giorgi's descriptive phenomenology, which elucidates the meanings of the phenomenon from the life experiences of participants. This approach is chosen for its ability to help grasp the beliefs of various actors involved in the study's phenomenon, conducting a phenomenological analysis using their own words [22, 23].

Qualitative data will be collected through focus groups. One set of groups will comprise primary care professionals, and another set will consist of patients with chronic conditions (conditions that are not transmitted from person to person, are long-lasting and generally progress slowly) [24] from primary care centers of different regions in Spain. Focus groups will be selected for their capacity to generate in-depth information regarding perceptions and opinions of the phenomenon through participant interaction, thus exploring a wide range of viewpoints on the subject [25]. To thoroughly analyse the study's phenomenon, intentional sampling of homogeneous groups will be employed, including both primary care professionals and patients with chronic conditions. Homogeneity within groups and heterogeneity across groups will be ensured to facilitate participants' free expression.

To organize the primary care professionals' focus groups, the following inclusion criteria will be employed: adult subjects (over 18 years of age), gender (male, female), area of expertise (medicine, nursing, physiotherapy), and years of professional experience ($<10$, 10–20, $>20$). For the selection of patients with chronic conditions, the following inclusion criteria will be used: subjects with chronic diseases, gender (male, female), employment status (retired, employed, unemployed, student), socioeconomic level (a socioeconomic level index was calculated with the items that referred to education level and occupation, classifying the socioeconomic level as: low, medium low, medium, medium high and high according to the scale proposed by the Spanish Society of Epidemiology) [26], education level (higher education or intermediate degree, high school graduate, or no education), and physical status (assessed using the SF-12 scale) (Appendix 1).

Expert qualitative methodologists will conduct the focus groups, following a common protocol encompassing data collection methodology and topic guidelines. The areas included in the topic guidelines were the most prevalent chronic diseases in Primary Care (ischemic heart disease, Alzheimer's disease and other dementias, cerebrovascular accident, cancer, chronic obstructive pulmonary disease, kidney disease, hypertension and diabetes mellitus) [27] to explore the barriers and facilitators to physical exercise prescription by primary care professionals in patients with chronic pathologies, and to explore patient barriers and facilitators and how these chronic pathologies influence compliance with physical exercise prescription. Each focus group will include a moderator (with the topic guidelines, leading the group, and posing questions) and another researcher as an observer. The focus groups will be conducted in a neutral, comfortable, noise-free, and private environment, lasting between 60 and 120 minutes. They will be audio-recorded with participants' permission and will include 5 to 8 participants per group.

A minimum of two focus groups will be conducted per cluster, i.e. for primary care professionals and for patients, but the final number will be determined by the data saturation criterion, which is reached when no novel analytical information is obtained [28].

Participants will be accessed through lists from health centers in the city where the study will be conducted. Additionally, study information will be disseminated through social and healthcare centers and media outlets to reach potential participants.

Data collection and analysis will follow an iterative circular process, where data collected in each focus group will refine the topic guidelines for subsequent groups. All data will be compared using the constant comparison method.

As data verification strategies, focus groups will be transcribed verbatim and then anonymized for analysis. Transcriptions will be returned to participants for agreement verification.

During the analysis phase, data will be grouped into themes and subthemes following Giorgi's phenomenological approach: i) collect and describe phenomenological data, ii) read the complete description in transcribed texts, iii) break descriptions into meaning units that are as descriptive as possible, avoiding premature interpretation, iv) group units by common meanings, forming clusters of meanings, and v) interpret clusters of meaning and identify themes that reveal the phenomenon's significance [22].

Two expert qualitative researchers will independently analyse the data, subsequently reaching consensus on results; a third investigator will mediate in case of disagreements. Data triangulation adopted through the application and combination of various data sources, research methods and theoretical frameworks, and carried out by three researchers, will provide diverse perspectives and deeper analysis, enhancing the findings' validity. Atlas-ti 9.0 software will assist in this phase.

Guba and Lincoln's criteria of credibility, transferability, dependability, and confirmability will be followed to ensure study reliability [29]. Additionally, the study will adhere to the Consolidated Criteria for Reporting Qualitative Research (COREQ) for qualitative studies [30].

## CPG development

The sequential process for developing the CPG for synthesizing the available scientific evidence on the effectiveness of physical exercise prescription in the most prevalent chronic conditions will follow the following outline [31]:

1. The CPG approach will require the involvement of healthcare professionals and graduates in physical activity and sports with experience in exercise and chronic diseases, ensuring [32]:

**Table 1. Formulation of clinical questions according to the PICO strategy.**

| Patient | Intervention | Comparison | Outcome* |
|---|---|---|---|
| *Type of population concerned (age, sex, stage of disease, comorbidity, etc.)* | *Pharmacological or nonpharmacological treatments, a diagnostic test, exposure to a risk factor,* | *Intervention to be compared with* | *Outcome variables of interest* |
| Adult subjects without pathology (prevention) or with pathology/comorbidity (treatment) | Physical exercise | intervention/Other intervention | Alzheimer's disease and other dementias<br>Stroke<br>Cancer<br>Chronic obstructive pulmonary disease<br>Kidney disease<br>High blood pressure<br>Diabetes mellitus |

*The most prevalent chronic diseases in primary care.

- Representation on the expert panel for the development of the CPG of professionals and patients with chronic diseases involved.

- Recognition of key aspects and possible barriers to future CPG implementation.

- Evaluation of available scientific evidence. Furthermore, it is important to consider involving patients in the CPG development process, from a literature review on patients' views on the CPG topic to incorporating patients into the development group [33]. Additionally, it is necessary for all individuals participating in the CPG development declare that there are no conflicts of interest.

2. Formulation of Clinical Questions: The goal of this section is to create a "question map," which will constitute the backbone of the CPG content [34]. First, a list of generic questions will be prepared to address the objectives set in the CPG. Second, questions will be reformulated specifically following the PICO strategy (Patient, Intervention, Comparison, and Outcome) (Table 1).

 The choice of outcome variables is crucial for the subsequent steps in CPG development. The use of the GRADE system involves an ordinal scale from 1 to 9 to classify outcome variables as critical or key for decision-making (7 to 9 points), important but not key for decision-making (4 to 6 points), and unimportant (1 to 3 points). In the case of the main outcome variables included in the CPG development (i.e., the most prevalent chronic pathologies in primary care), quality of life, comorbidity, and mortality associated with these chronic diseases stand out as final outcome variables [35]. Some examples of generic questions for this CPG development would be: Health education in adult subjects without pathology, is health education on physical activity/exercise provided by professionals? For prevention in adult subjects with personal history and/or specific risk factors, should physical exercise be considered to reduce disease risk? Regarding nonpharmacological treatment and monitoring in adult subjects with chronic pathology, should physical exercise be recommended to improve their quality of life? What type of exercise program is most effective for improving clinical and functional parameters? Morbidity in adult subjects with chronic pathology, Does physical exercise reduce morbidity associated with the pathology?

3. Search and Selection of Evidence: Clinical questions form the basis of the literature review since their components determine the inclusion criteria for studies [36]. Two principles are fundamental in the evidence search:

- The most appropriate databases (such as SPORTDiscus, Medline, CINAHL, and Embase) were selected.

- Choice of the most appropriate study design for the clinical question. In this case, to evaluate the effectiveness of a physical exercise intervention in preventing and treating different chronic pathologies, the preferred study design would be systematic reviews and/or meta-analyses of randomized controlled trials and, if unavailable, randomized controlled trials themselves. A description of the work plan will be carried out, including the databases to be used, the time period, and the languages that will delimit the search. A preliminary search of the literature is important to gain perspective on the aspects that the CPG should cover, helping to identify and define key areas, as well as offering an image of the volume of literature and the work that its review may generate.

4. Evaluation and Synthesis of Evidence:

   Evidence-based medicine is driving transformations in the way information is evaluated and synthesized, as well as in the formulation of recommendations in the development of CPG, promoting transparency in this process [37]. The GRADE system of evidence evaluation and formulation of recommendations is the latest methodological evolution in the design of CPGs, which proposes a series of sequential judgments in the formulation of recommendations incorporating not only the quality of the evidence but also the balance between benefits and risks, patients' beliefs and economic cost.

5. Formulation of Recommendations: To structure the information and provide the RedExAP study decisions with maximum transparency, the EtD (Evidence to Decision frameworks) proposal within the GRADE system will be used [38]. This framework is structured into three fundamental parts:

- Formulation of the question.

- Evaluation of criteria.

- Conclusions. Preparing these frameworks requires the systematic review of the available evidence for different aspects addressing both the effectiveness of physical exercise on the most prevalent chronic diseases and aspects of patient outcome assessment, resource use and costs, acceptability or feasibility, among others. To carry out the implementability of the CPG, it is necessary to evaluate the clarity of presentation, in terms of language, structure, and format aspects, as well as the applicability of the CPG, referring to possible barriers and facilitators for its implementation, strategies to improve its adoption, and the implications of applying the CPG to resources. External review is a fundamental stage to refine and enrich the CPG [39]. Multidisciplinary professionals and external patients will be invited to evaluate the CPG. The objective of this type of review is to increase the external validity of the CPG and its recommendations and promote a final product more in line with the environment it is intended for. It also aims to encourage the acceptance of the CPG by end-users, thus facilitating its implementation.

## Ethical considerations

This study protocol was approved by the Clinical Research Ethics Committee of the Hospital Virgen de la Luz in the city of Cuenca where the study is to be performed (REG: 2023/PI1723).

Participants will be informed of the objectives and methods of the study and will be asked to confirm in writing their approval to participate in the study.

## Discussion

The results of the RedExAP study aim to improve the health of people suffering from the most prevalent chronic diseases in primary care by prescribing personalized physical activity [40]. Chronic diseases are closely related to lifestyles; therefore, a CPG will be developed that summarizes the highest-quality scientific evidence available to address the doubts of healthcare professionals when prescribing personalized physical activity [41]. This CPG will have a transdisciplinary approach, in which all health professionals will work towards improving people's health, assuming that there are areas of competence that go beyond each discipline and require the knowledge of a multidisciplinary team, where the knowledge of all disciplines complements each other [42]. Furthermore, meeting the aims of the RedExAP study could highlight new social needs and the fundamental role of physical educators professionals specialized in prescribing, designing, and supervising physical exercise within the primary care team for the benefit of the population with chronic diseases [43]. Physical educators integrated into the primary care team must play a key role in creating spaces for supervised physical exercise in primary care centers, helping to improve the alarming data on physical inactivity and the expected negative projections. These changes would result in better population health and greater social well-being, a key goal of the 2030 Agenda, which is based on the principles of sustainability, equity, inclusion and respect for human rights, promoting health and the wellbeing of the population where the promotion of physical exercise is a fundamental support [44]. The strategy for disseminating the RedExAP research results is a priority, as it is intended for the results to have national and international visibility to impact the improvement of society's health.

The relevance and originality of the current study primarily lie in the three pillars on which it is based: transdisciplinarity, personalized prescription, and social determinants of health [45]. The social determinants of health refer to environmental factors that affect health, such as gender, marital status, ethnicity, occupation, educational level, and socioeconomic status, among others [46, 47]. There is evidence linking these unfavourable social conditions with adverse health outcomes such as diabetes mellitus, hypertension, heart failure, stroke, and mortality [48–51], thus considering these determinants as indicators of cardiovascular risk [52]. There is more than ample evidence that physical exercise is one of the most effective treatments for chronic diseases responsible for most deaths worldwide [53]. Not offering a treatment whose effectiveness is so evident, scientifically speaking, approaches medical negligence. Currently, we are experiencing an unprecedented health crisis in which primary care and society suffer due to the lack of healthcare personnel [54]. Transdisciplinarity, a principle of unity of theoretical and practical knowledge beyond disciplines, could provide an important solution to this problem [55]. The contribution of physical educators in healthcare teams and centers would reduce the care burden not only by efficiently addressing pathologies that can benefit greatly but also by preventing other comorbidities and even the pathologies themselves, relieving other professionals' work. The study is original because, to the best of our knowledge, there is no other study in the world that has made the relationship between physical exercise, primary care, and health professionals its main objective, with such a clear approach based on transdisciplinarity and social determinants of health. Additionally, the study aims to promote health promotion as an essential part of the primary care strategy, an approach that emerged from the health system's approach from the 1980s and which continues to be an important component of primary health care today. The current study can be considered a revival of the approach that was so successful at that time. Primary care professionals should prescribe exercise based on scientific evidence; however, for this purpose, they need evidence that is summarized and translated into understandable language, which allows them to apply it to the

patient's real world. The study is innovative because it offers a transdisciplinary solution to a global problem that can improve the health of populations and the cost-effectiveness of health systems [56].

## Supporting information

**S1 Appendix. SF-12 health questionnaire.**
(PDF)

## Author Contributions

**Conceptualization:** Iván Cavero-Redondo.

**Data curation:** José Ignacio Recio-Rodriguez, Iván Cavero-Redondo.

**Formal analysis:** Alicia Saz-Lara, Iván Cavero-Redondo.

**Investigation:** José Alberto Martínez Hortelano.

**Methodology:** Alicia Saz-Lara, Iván Cavero-Redondo.

**Resources:** María Medrano, Miriam Garrido Miguel, Montserrat García-Sastre.

**Software:** María Medrano, Raquel Luengo-González.

**Supervision:** Iván Cavero-Redondo.

**Validation:** Miriam Garrido Miguel, Montserrat García-Sastre, José Ignacio Recio-Rodriguez, Daniel Lozano-Cuesta.

**Visualization:** Raquel Luengo-González.

**Writing – original draft:** Alicia Saz-Lara, José Alberto Martínez Hortelano.

**Writing – review & editing:** José Ignacio Recio-Rodriguez, Daniel Lozano-Cuesta.

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
