## [Decision Letter · Decision Letter 0]

19 Jan 2024

PONE-D-23-38977Exercise research for the prevention and treatment of chronic diseases in primary care: protocol for the rationale of the RedExAP studyPLOS ONE

Dear Dr. Martínez Hortelano,

Thank you for submitting your manuscript to PLOS ONE. After careful consideration, we feel that it has merit but does not fully meet PLOS ONE’s publication criteria as it currently stands. Therefore, we invite you to submit a revised version of the manuscript that addresses the points raised during the review process.

We look forward to receiving your revised manuscript.

Kind regards,

Hidetaka Hamasaki

Academic Editor

PLOS ONE

Journal Requirements:

Reviewers' comments:

Reviewer's Responses to Questions

**Comments to the Author**

1. Does the manuscript provide a valid rationale for the proposed study, with clearly identified and justified research questions?

Reviewer #1: No

Reviewer #2: Yes

2. Is the protocol technically sound and planned in a manner that will lead to a meaningful outcome and allow testing the stated hypotheses?

Reviewer #1: Yes

Reviewer #2: Partly

3. Is the methodology feasible and described in sufficient detail to allow the work to be replicable?

Reviewer #1: Yes

Reviewer #2: Yes

4. Have the authors described where all data underlying the findings will be made available when the study is complete?

Reviewer #1: Yes

Reviewer #2: Yes

5. Is the manuscript presented in an intelligible fashion and written in standard English?

Reviewer #1: Yes

Reviewer #2: Yes

6. Review Comments to the Author

You may also provide optional suggestions and comments to authors that they might find helpful in planning their study.

Reviewer #1: Comments to the authors

I would like to congratulate the authors on a well compiled piece of work, including the basic steps of generating a guideline, which I find to be very relevant to patients, health workers and society in general, not to forget the economy behind treating chronic diseases.

I find the protocol well written, clear and concise.

I do however find some the references to be a bit “old”, and there are more recent studies on the subject than those included in the study. This is not a fault as such, but hopefully some of the newer studies will be found in the future systematic review intended as part of the upcoming study.

I have as such no comments to the authors, except I hope all the best for the coming projects. There is a large amount of work ahead, and I hope to see the result in time forthcoming.

Reviewer #2: Title:

PONE-D-23-38977: Exercise research for the prevention and treatment of chronic diseases in primary care: protocol for the rationale of the RedExAP study.

General comments:

Thank you for giving me the opportunity to review this manuscript. It is an interesting manuscript describing a research protocol consisting of two phases of study i.e. a qualitative study and clinical practice guidelines (CPG) development. However, further details are required to add clarity to the manuscript.

Specific comments:

Title:

1. ‘Exercise research’ is rather vague and can be improved to reflect more specifically what this exercise research is about.

Introduction:

2. The first and second paragraphs describe the benefits / effectiveness of exercise versus pharmacological intervention. However, it is not stated what drugs / what condition or disease(s) were previously studied. This is in consideration that in many diseases, medications have specific function in treatment. It may be difficult to compare head-to-head specific medication with exercise especially looking at mortality outcome.

3. The third paragraph: ‘….physical exercise has proven to be superior to the pharmacological treatments commonly used in clinical practice in some cases…’ Suggest to provide examples of diseases to explain ‘some cases’.

4. The term ‘rehabilitator’ – does it mean rehabilitation medicine physicians only or includes other rehabilitation therapist?

5. This sentence need reference: ‘….health services have incorporated the prescription of physical activity into the range of services they offer, some evidence shows that either this service does not reach users or is provided inadequately.’

6. The fifth paragraph describes ‘the urgency for this evidenced scientific knowledge to be accessible to healthcare professionals.’ Suggest adding references to support the statement that knowledge on exercise is not accessible.

7. Aim number (iii) is not clear. The aim is to synthesize evidence on the effectiveness of physical exercise prescription. This is more appropriately conducted through a systematic review.

Material and methods:

8. First line – qualitative study is to explore rather than analyse the barriers and facilitators.

Will the results obtained in the qualitative study contribute to the development of the CPG?

Qualitative study

9. Need to include operative definitions of terms used in this research: ‘primary care professionals’, ‘patients with chronic conditions’, ‘primary care centres.’

10. The description of ‘criteria’ for selecting primary care professionals or patients is confusing. Does it mean the inclusion criteria or the different groups for intentional (purposive) sampling? Therefore, need to be clarified.

11. What are the criteria to decide a participant to be of low, medium or high socioeconomic level?

12. Suggest including a short description of SF-12 scale or include it as an appendix.

13. A short description of how the topic guideline is developed should be included.

14. The regions or location of the health/primary care centres involved in participant recruitment should be included. Are they rural or urban areas?

15. ‘Data triangulation by three investigators’ need to be clarified – how will it be done?

CPG development

16. Refer to comment (7) regarding the appropriate method of synthesizing scientific evidence on effectiveness of physical exercise prescription. The general aim of CPG development is to provide an evidence-based guide/recommendations for health care providers to improve quality of health care. The authors need to clarify the aims. ‘Most prevalent chronic conditions’ need to be defined.

17. Point (1) following the first paragraph is confusing. Does it mean that healthcare professionals and patients will be the expert panel in the CPG development? What is meant by ‘all individuals need to declare their interests?’

18. Table 1 – the outcomes in this table are different from what is stated in the paragraph below the table 1 i.e. quality of life etc.

19. The generic question of ‘is health education on physical activity/exercises provided by professionals’ need clarification as it does not seem to align with the CPG’s aims.

20. The sentence ‘The application of evidence-based medicine is responsible for the current changes….’ Is not clear and would benefit from a re-structuring of the sentence.

Discussion:

21. Need a definition for ‘physical educators.’

22. It is stated that this research ‘…will highlight new social needs…’ however, this is not one of the study’s aim.

23. The second paragraph on economic of physical inactivity is more relevant as part of introduction to the study.

24. Page 21 – need reference to support the statement regarding health crisis due to lack of healthcare personnel. Furthermore, it does not reflect the aim of this study.

25. ‘Social determinants of health’ need further elaboration.

26. The statement; ‘study aims to recover the health system's approach from the 1980s when health promotion was an essential part of the primary care strategy’ needs clarification. Is it a specific region’s health system approach? World Health Organization still highlights health promotion as a component of primary health care (not just in the 1980s). Refer to: https://www.who.int/health-topics/primary-health-care#tab=tab_1

7. PLOS authors have the option to publish the peer review history of their article (what does this mean?). If published, this will include your full peer review and any attached files.

Reviewer #1: No

Reviewer #2: No

---

## [Author Response · Author response to Decision Letter 0]

26 Feb 2024

Reviewer 1

Comments to the authors

I would like to congratulate the authors on a well compiled piece of work, including the basic steps of generating a guideline, which I find to be very relevant to patients, health workers and society in general, not to forget the economy behind treating chronic diseases.

I find the protocol well written, clear and concise.

I do however find some the references to be a bit “old”, and there are more recent studies on the subject than those included in the study. This is not a fault as such, but hopefully some of the newer studies will be found in the future systematic review intended as part of the upcoming study.

I have as such no comments to the authors, except I hope all the best for the coming projects. There is a large amount of work ahead, and I hope to see the result in time forthcoming.

Authors: 

The reviewer´s comment seems judicious. As suggested, we will include more recent references on the subject in the future systematic review. We greatly appreciate the reviewer´s time in reviewing the manuscript. 

Reviewer 2

Title:

PONE-D-23-38977: Exercise research for the prevention and treatment of chronic diseases in primary care: protocol for the rationale of the RedExAP study.

General comments:

Thank you for giving me the opportunity to review this manuscript. It is an interesting manuscript describing a research protocol consisting of two phases of study i.e. a qualitative study and clinical practice guidelines (CPG) development. However, further details are required to add clarity to the manuscript. 

Authors: 

Thank you for the reviewer’s comment. We greatly appreciate the reviewer´s time in reviewing the manuscript.

Specific comments:

Title: 

1. ‘Exercise research’ is rather vague and can be improved to reflect more specifically what this exercise research is about. 

Authors: 

Thank you for the reviewer’s comment. As suggested, we have modified the title as follows: “Effect of exercise for the prevention and treatment of chronic diseases in primary care: rationale of the RedExAP study”.

Introduction:

2. The first and second paragraphs describe the benefits / effectiveness of exercise versus pharmacological intervention. However, it is not stated what drugs / what condition or disease(s) were previously studied. This is in consideration that in many diseases, medications have specific function in treatment. It may be difficult to compare head-to-head specific medication with exercise especially looking at mortality outcome. 

Authors: 

Thank you for the reviewer’s comment. As suggested, we have included information about the effectiveness of physical exercise and pharmacological interventions in different diseases.

“[…] Although it is true that there is a lack of evidence demonstrating the effectiveness of both physical exercise and pharmacological interventions in the prevention of coronary heart disease and diabetes mellitus, it has been shown that physical exercise could be effective in the treatment of patients with stroke, while treatment with diuretics was more effective than exercise in patients with heart failure (Naci and Ioannidis.; 2013). Furthermore, there is evidence showing the effectiveness of physical exercise interventions to improve cognition in patients with cognitive impairment and Alzheimer's disease (Chen et al.; 2018;).”

3. The third paragraph: ‘….physical exercise has proven to be superior to the pharmacological treatments commonly used in clinical practice in some cases…’ Suggest to provide examples of diseases to explain ‘some cases’. 

Authors: 

Thank you for the reviewer’s comment. As suggested, we have provided examples of diseases to explain some cases. 

“In recent years, numerous published studies have assessed the effectiveness of physical exercise in the preventive or therapeutic management of many prevalent chronic conditions that affect children (Janssen and Leblanc.; 2010; Ortega et al.; 2008), adults and the elderly (Pedersen and Saltin.; 2015), and, although the evidence is limited, physical exercise has proven to be superior or potentially similar to the pharmacological treatments commonly used in clinical practice in some cases, such as in patients with stroke, heart failure or Diabetes Mellitus, among other (Naci and Ioannidis.; 2013); however, the reasons why this evidence does not translate into clinical practice remain to be elucidated.”

4. The term ‘rehabilitator’ – does it mean rehabilitation medicine physicians only or includes other rehabilitation therapist?

Authors: 

Thank you for the reviewer’s comment. As suggested, we have clarified this issue as follows: 

“A transdisciplinary approach is needed to include physical activity and sports in health care systems, so the importance of including different specialist healthcare professional profiles (such as cardiologists, geriatricians, rehabilitation medicine physicians, and physiotherapists), primary care physicians and nurses, and physical activity and sports graduates could play a key role in integrating exercise prescription, design, and intervention implementation to prevent and manage chronic diseases. […]”

5. This sentence need reference: ‘….health services have incorporated the prescription of physical activity into the range of services they offer, some evidence shows that either this service does not reach users or is provided inadequately.’ 

Authors: 

Thank you for the reviewer’s comment. As suggested, we have included a reference for the previous sentence. 

“[…] Therefore, although theoretically, health services have incorporated the prescription of physical activity into the range of services they offer, some evidence shows that either this service does not reach users or is provided inadequately (Müller-Riemenschneider.; 2008).”

In the Reference section: 

“Müller-Riemenschneider F, Reinhold T, Nocon M, Willich SN. Long-term effectiveness of interventions promoting physical activity: a systematic review. Prev Med. 2008; 47(4): 354-68.”

6. The fifth paragraph describes ‘the urgency for this evidenced scientific knowledge to be accessible to healthcare professionals.’ Suggest adding references to support the statement that knowledge on exercise is not accessible. 

Authors: 

Thank you for the reviewer’s comment. As suggested, we have included a reference for the previous sentence. 

In the Reference section: 

“Lobelo F, Rohm Young D, Sallis R, Garber MD, Billinger SA, Duperly J, et al. Routine Assessment and Promotion of Physical Activity in Healthcare Settings: A Scientific Statement From the American Heart Association. Circulation. 2018; 137(18): e495-e522.”

7. Aim number (iii) is not clear. The aim is to synthesize evidence on the effectiveness of physical exercise prescription. This is more appropriately conducted through a systematic review. 

Authors: 

Thank you for the reviewer’s comment. As suggested, we have clarified the (iii) aim as follows: 

“[…] iii) synthesize, through a clinical practice guideline (CPG) or white paper, the recommendations on the effectiveness of physical exercise prescription in the prevention and treatment of the most prevalent chronic conditions in Western countries. […]”

Material and methods:

8. First line – qualitative study is to explore rather than analyse the barriers and facilitators. 

Will the results obtained in the qualitative study contribute to the development of the CPG?

Authors:

Thank you for the reviewer’s comment. As suggested, we have modified this issue as follows:

“The RedExAP study will be a qualitative study to analyse barriers and facilitators for physical exercise prescription and the results obtained in this study will contribute to the development of a CPG for the effectiveness of physical exercise prescription in the most prevalent chronic conditions.”

Qualitative study

9. Need to include operative definitions of terms used in this research: ‘primary care professionals’, ‘patients with chronic conditions’, ‘primary care centres.’ 

Authors: 

Thank you for the reviewer’s comment. As suggested, we have included the definitions of ‘primary care professionals’, ‘patients with chronic conditions’, and ‘primary care centres’. 

“A qualitative study will be conducted to comprehend the challenges faced by primary care professionals (health professionals, such as physician and nurses, among others, who care for people as a whole, in the context of the patient's clinical history and their life circumstances) in prescribing physical exercise to patients and understand the difficulties perceived by patients at primary care centers (health services that satisfy the needs of basic care, and that guarantee comprehensive quality care, from promotion and prevention to treatment, rehabilitation and palliative care) in adhering to received physical activity and exercise recommendations (Shi.; 2012). […]”

“Qualitative data will be collected through focus groups. One set of groups will comprise primary care professionals, and another set will consist of patients with chronic conditions (conditions that are not transmitted from person to person, are long-lasting and generally progress slowly) (Sevick et al.; 2007) from primary care centers of different regions in Spain. […]”

In the References section: 

‐ Shi L. The impact of Primary Care: a focused review. Scientifica. 2012; 2012: 432892.

‐ Sevick MA, Trauth JM, Ling BS, Anderson RT, Piatt GA, Kilbourne AM, et al. Patients with Complex Chronic Diseases: perspectives on supporting self-management. J Gen Intern Med. 2007; 22 Suppl 3(Suppl 3): 438-444.

10. The description of ‘criteria’ for selecting primary care professionals or patients is confusing. Does it mean the inclusion criteria or the different groups for intentional (purposive) sampling? Therefore, need to be clarified. 

Authors: 

The reviewer´s comment seems judicious. As suggested, we have clarified this issue as follows: 

“To organize the primary care professionals' focus groups, the following inclusion criteria will be employed: gender (male, female), area of expertise (medicine, nursing, physiotherapy), and years of professional experience (<10, 10-20, >20). For the selection of patients with chronic conditions, the following inclusion criteria will be used: gender (male, female), employment status (retired, employed, unemployed, student), socioeconomic level (low, medium, high), education level (higher education or intermediate degree, high school graduate, or no education), and physical status (assessed using the SF-12 scale).”

11. What are the criteria to decide a participant to be of low, medium or high socioeconomic level?

Authors: 

The reviewer´s comment seems judicious. As suggested, we have included the criteria to classify the degrees of socioeconomic level. 

“[…] socioeconomic level (a socioeconomic level index was calculated with the items that referred to education level and occupation, classifying the socioeconomic level as: low, medium low, medium, medium high and high according to the scale proposed by the Spanish Society of Epidemiology) (Domingo-Salvany et al.; 2000), […]”

In the Reference section:

- “Domingo-Salvany A, Regidor E, Alonso J, Alvarez-Dardet C. Proposal for a social class measure. Working Group of the Spanish Society of Epidemiology and the Spanish Society of Family and Community Medicine. Atencion Primaria/Sociedad Española De Medicina De Familia y Comunitaria. 2000; 25(5): 350.”

12. Suggest including a short description of SF-12 scale or include it as an appendix.

Authors: 

Thank you for the reviewer's comment. As suggested, we have included a description of the SF-12 scale as an appendix.

“[…] and physical status (assessed using the SF-12 scale) (Appendix 1).”

13. A short description of how the topic guideline is developed should be included.

Authors: 

The reviewer´s comment seems judicious. As suggested, we have included a description of the topic guidelines. 

“[…] The areas included in the topic guidelines were the most prevalent chronic diseases in Primary Care (ischemic heart disease, Alzheimer's disease and other dementias, cerebrovascular accident, cancer, chronic obstructive pulmonary disease, kidney disease, hypertension and Diabetes Mellitus). […]”

In the References section:

Hajat C and Stein E. The global burden of multiple chronic conditions: A narrative review. Prev Med Rep. 2018; 12: 284-293. 

14. The regions or location of the health/primary care centres involved in participant recruitment should be included. Are they rural or urban areas?

Authors: 

Thank you for the reviewer’s comment. As suggested, we have included the location of the health/primary care centers involved in participant recruitment.

“Participants will be accessed through lists from health centers in the city where the study will be conducted. […]”

15. ‘Data triangulation by three investigators’ need to be clarified – how will it be done?

Authors: 

The reviewer´s comment seems judicious. As suggested, we have clarified this issue as follows: 

“[…] Data triangulation adopted through the application and combination of various data sources, research methods and theoretical frameworks, and carried out by three researchers, will provide diverse perspectives and deeper analysis, enhancing the findings' validity. […]”

CPG development

16. Refer to comment (7) regarding the appropriate method of synthesizing scientific evidence on effectiveness of physical exercise prescription. The general aim of CPG development is to provide an evidence-based guide/recommendations for health care providers to improve quality of health care. The authors need to clarify the aims. ‘Most prevalent chronic conditions’ need to be defined. 

Authors: 

Thank you for the reviewer’s comment. As suggested, we have clarified the aim and defined the most prevalent chronic diseases.

“[…] iii) synthesize, through a clinical practice guideline (CPG) or white paper, the recommendations on the effectiveness of physical exercise prescription in the prevention and treatment of the most prevalent chronic conditions in Western countries. […]”

“[…] The areas included in the topic guidelines were the most prevalent chronic diseases in Primary Care (ischemic heart disease, Alzheimer's disease and other dementias, cerebrovascular accident, cancer, chronic obstructive pulmonary disease, kidney disease, hypertension and Diabetes Mellitus) (Hajat and Stein,; 2018). […]”

In the References section: 

‐ “Hajat C and Stein E. T global burden of multiple chronic conditions: A narrative review. Prev Med Rep. 2018; 12: 284-293.”

17. Point (1) following the first paragraph is confusing. Does it mean that healthcare professionals and patients will be the expert panel in the CPG development? What is meant by ‘all individuals need to declare their interests?’.

Authors: 

Thank you for the reviewer’s comment. As suggested, we have clarified these issues as follows:

“Representation on the expert panel for the development of the CPG of professionals and patients with chronic diseases involved.”

“[…] Additionally, it is necessary for all individuals participating in the CPG development declare that there are no conflicts of interest.”

18. Table 1 – the outcomes in this table are different from what is stated in the paragraph below the table 1 i.e. quality of life etc. 

Authors: 

The reviewer´s comment seems judicious. As suggested, we have clarified this issue as follows: 

“[…] In the case of the main outcome variables included in the CPG development (i.e., the most prevalent chronic pathologies in primary care), quality of life, comorbidity, and mortality associated with these chronic diseases stand out as final outcome variables (Guyatt et al.; 2011). […]”

19. The generic question of ‘is health education on physical activity/exercises provided by professionals’ need clarification as it does not seem to align with the CPG’s aims.

Authors: 

The reviewer´s comment seems judicious. As suggested, we have clarified this issue as follows: 

“[…] iii) synthesize, through a clinical practice guideline (CPG) or white paper, the recommendations on the effectiveness of physical exercise prescription in the prevention and treatment of the most prevalent chronic conditions in 

---

## [Decision Letter · Decision Letter 1]

13 Mar 2024

PONE-D-23-38977R1Effect of exercise for the prevention and treatment of chronic diseases in primary care: rationale of the RedExAP studyPLOS ONE

Dear Dr. Martínez Hortelano,

Thank you for submitting your manuscript to PLOS ONE. After careful consideration, we feel that it has merit but does not fully meet PLOS ONE’s publication criteria as it currently stands. Therefore, we invite you to submit a revised version of the manuscript that addresses the points raised during the review process.

We look forward to receiving your revised manuscript.

Kind regards,

Hidetaka Hamasaki

Academic Editor

PLOS ONE

Journal Requirements:

Additional Editor Comments:==============================

**ACADEMIC EDITOR:**

*Comments from PLOS editorial office: When you resubmit, please modify the title of your submission to include the words "study protocol".*

Reviewers' comments:

Reviewer's Responses to Questions

**Comments to the Author**

1. Does the manuscript provide a valid rationale for the proposed study, with clearly identified and justified research questions?

Reviewer #2: Yes

2. Is the protocol technically sound and planned in a manner that will lead to a meaningful outcome and allow testing the stated hypotheses?

Reviewer #2: Yes

3. Is the methodology feasible and described in sufficient detail to allow the work to be replicable?

Reviewer #2: Yes

4. Have the authors described where all data underlying the findings will be made available when the study is complete?

Reviewer #2: No

5. Is the manuscript presented in an intelligible fashion and written in standard English?

Reviewer #2: Yes

6. Review Comments to the Author

You may also provide optional suggestions and comments to authors that they might find helpful in planning their study.

Reviewer #2: Title:

PONE-D-23-38977-R1: Effect of exercise for the prevention and treatment of chronic diseases in primary care: rationale of the RedExAP study.

General comments:

The authors have addressed the previous comments. However, there are a few further comments and clarifications required as below.

Specific comments:

Title:

The aims of this study are to explore barriers and facilitators for physical exercise prescription and developing a CPG for effectiveness of physical exercise prescription (and not to determine the effect of exercise). The authors may consider the title as ‘Exercise prescription for the prevention and treatment of chronic diseases.…., instead of ‘effect of exercise’ as more appropriate in reflecting the study.

Material and methods:

Qualitative study

In the paragraph on inclusion criteria, the authors described the socio-demographic information of the respondents that is planned to be collected and how it is to be categorized (which is also important in the methods section). Inclusion criteria by definition is which type of population can participate in the study.

Page 45, first paragraph – a topic guide would include questions to ask the participants during the focus group discussion (as a guide for the focus group moderator). To ask the participants regarding the diseases listed does not seem to align with the aim of exploring barriers and facilitators for physical exercise prescription.

Page 45 – ‘A minimum of two focus groups will be conducted per segment’, it is not clear what constitutes ‘segment’.

Page 45 – ‘Data collection and analysis will follow an interactive circular process’ – the word ‘interactive’ should be ‘iterative’.

Discussion:

Not all readers are familiar with 2030 Agenda, therefore, the authors can be more specific on this agenda. Similarly, 2030 Agenda is stated in the abstract.

Others:

‘Diabetes mellitus’ should be in lower case.

‘CINHAL’ – the correct spelling is CINAHL.

There are missing information in the revised main manuscript as below:

i. Ethical consideration

ii. Funding details

iii. Ethics approval and consent to participate.

7. PLOS authors have the option to publish the peer review history of their article (what does this mean?). If published, this will include your full peer review and any attached files.

Reviewer #2: No

---

## [Author Response · Author response to Decision Letter 1]

24 Mar 2024

Additional Editor Comments:

ACADEMIC EDITOR:

Comments from PLOS editorial office: When you resubmit, please modify the title of your submission to include the words "study protocol".

Authors:

Thank you for the reviewer’s comment. We greatly appreciate the reviewer´s time in reviewing the manuscript. As suggested, we have modified the title of the manuscript as follows: 

“Exercise prescription for the prevention and treatment of chronic diseases in primary care: protocol of the RedExAP study.”

REVIEWER 2:

Title:

PONE-D-23-38977-R1: Effect of exercise for the prevention and treatment of chronic diseases in primary care: rationale of the RedExAP study.

General comments:

The authors have addressed the previous comments. However, there are a few further comments and clarifications required as below. 

Authors:

Thank you for the reviewer’s comment. We greatly appreciate the reviewer´s time in reviewing the manuscript.

Specific comments:

Title: 

The aims of this study are to explore barriers and facilitators for physical exercise prescription and developing a CPG for effectiveness of physical exercise prescription (and not to determine the effect of exercise). The authors may consider the title as ‘Exercise prescription for the prevention and treatment of chronic diseases.…., instead of ‘effect of exercise’ as more appropriate in reflecting the study.

Authors:

Thank you for the reviewer’s comment. As suggested, we have modified the title of the manuscript as follows: 

“Exercise prescription for the prevention and treatment of chronic diseases in primary care: protocol of the RedExAP study.”

Material and methods:

Qualitative study

In the paragraph on inclusion criteria, the authors described the socio-demographic information of the respondents that is planned to be collected and how it is to be categorized (which is also important in the methods section). Inclusion criteria by definition is which type of population can participate in the study. 

Authors:

The reviewer´s comment seems judicious. As suggested, we have included the population type. 

“To organize the primary care professionals' focus groups, the following inclusion criteria will be employed: adult subjects (over 18 years of age), gender (male, female), area of expertise (medicine, nursing, physiotherapy), and years of professional experience (<10, 10-20, >20). For the selection of patients with chronic conditions, the following inclusion criteria will be used: healthy adult subjects or patients with chronic conditions (over 18 years of age), gender (male, female), employment status (retired, employed, unemployed, student), socioeconomic level (a socioeconomic level index was calculated with the items that referred to education level and occupation, classifying the socioeconomic level as: low, medium low, medium, medium high and high according to the scale proposed by the Spanish Society of Epidemiology) [26], education level (higher education or intermediate degree, high school graduate, or no education), and physical status (assessed using the SF-12 scale) (Appendix 1).”

Page 45, first paragraph – a topic guide would include questions to ask the participants during the focus group discussion (as a guide for the focus group moderator). To ask the participants regarding the diseases listed does not seem to align with the aim of exploring barriers and facilitators for physical exercise prescription. 

Authors:

Thank you for the reviewer’s comment. As suggested, we have explained this issue as follows:

“[…] The areas included in the topic guidelines were the most prevalent chronic diseases in Primary Care (ischemic heart disease, Alzheimer's disease and other dementias, cerebrovascular accident, cancer, chronic obstructive pulmonary disease, kidney disease, hypertension and diabetes mellitus) [27] to explore the barriers and facilitators to physical exercise prescription by primary care professionals in patients with chronic pathologies, and to explore patient barriers and facilitators and how these chronic pathologies influence compliance with physical exercise prescription. […]”

Page 45 – ‘A minimum of two focus groups will be conducted per segment’, it is not clear what constitutes ‘segment’.

Authors:

The reviewer´s comment seems judicious. As suggested, we have modified this term and have explained it. 

“A minimum of two focus groups will be conducted per group, i.e. for primary care professionals and for patients, but the final number will be determined by the data saturation criterion, which is reached when no novel analytical information is obtained [28].”

Page 45 – ‘Data collection and analysis will follow an interactive circular process’ – the word ‘interactive’ should be ‘iterative’. 

Authors:

Thank you for the reviewer’s comment. As suggested, we have replaced the word "interactive" with "iterative".

“Data collection and analysis will follow an iterative circular process, where data collected in each focus group will refine the topic guidelines for subsequent groups. All data will be compared using the constant comparison method.”

Discussion:

Not all readers are familiar with 2030 Agenda, therefore, the authors can be more specific on this agenda. Similarly, 2030 Agenda is stated in the abstract. 

Authors:

The reviewer´s comment seems judicious. As suggested, we have specified the main aims of 2020 Agenda and their relationship to physical exercise.

In the Abstract section:

“[…] The innovative transdisciplinary approach aligns with the 2030 Agenda, promoting better population health and greater social well-being, and shows promise in alleviating chronic disease burdens. […]”

In the Discussion section: 

“[…] These changes would result in better population health and greater social well-being, a key goal of the 2030 Agenda, which is based on the principles of sustainability, equity, inclusion and respect for human rights, promoting health and the well-being of the population where the promotion of physical exercise is a fundamental support [44]. […]”

Others: 

‘Diabetes mellitus’ should be in lower case. 

Authors:

Thank you for the reviewer’s comment. As suggested, we have included the term “diabetes mellitus” in lower case throughout the manuscript.

“In recent years, numerous published studies have assessed the effectiveness of physical exercise in the preventive or therapeutic management of many prevalent chronic conditions that affect children [5,6], adults and the elderly [7], and, although the evidence is limited, physical exercise has proven to be superior or potentially similar to the pharmacological treatments commonly used in clinical practice in some cases, such as in patients with stroke, heart failure or diabetes mellitus, among other [2]; […]”

“[…] The areas included in the topic guidelines were the most prevalent chronic diseases in Primary Care (ischemic heart disease, Alzheimer's disease and other dementias, cerebrovascular accident, cancer, chronic obstructive pulmonary disease, kidney disease, hypertension and diabetes mellitus) [27]. […]”

‘CINHAL’ – the correct spelling is CINAHL.

Authors:

Thank you for the reviewer’s comment. As suggested, we have modified the word CINHAL to CINAHL.

“The most appropriate databases (such as SPORTDiscus, Medline, CINAHL, and Embase) were selected.”

There are missing information in the revised main manuscript as below:

i. Ethical consideration

ii. Funding details

iii. Ethics approval and consent to participate.

Authors:

Thank you for the reviewer’s comment. Information on ethical consideration, funding details, and ethics approval and consent to participate are shown in the unblinded main manuscript.

---

## [Decision Letter · Decision Letter 2]

10 Apr 2024

Exercise prescription for the prevention and treatment of chronic diseases in primary care: protocol of the RedExAP study

PONE-D-23-38977R2

Dear Dr. Martínez Hortelano,

We’re pleased to inform you that your manuscript has been judged scientifically suitable for publication and will be formally accepted for publication once it meets all outstanding technical requirements.

Kind regards,

Hidetaka Hamasaki

Academic Editor

PLOS ONE

Additional Editor Comments (optional):

Reviewers' comments:

Reviewer's Responses to Questions

**Comments to the Author**

1. Does the manuscript provide a valid rationale for the proposed study, with clearly identified and justified research questions?

Reviewer #2: Yes

2. Is the protocol technically sound and planned in a manner that will lead to a meaningful outcome and allow testing the stated hypotheses?

Reviewer #2: Yes

3. Is the methodology feasible and described in sufficient detail to allow the work to be replicable?

Reviewer #2: Yes

4. Have the authors described where all data underlying the findings will be made available when the study is complete?

Reviewer #2: No

5. Is the manuscript presented in an intelligible fashion and written in standard English?

Reviewer #2: Yes

6. Review Comments to the Author

You may also provide optional suggestions and comments to authors that they might find helpful in planning their study.

Reviewer #2: PONE-D-23-38977R2

Title:

Exercise prescription for the prevention and treatment of chronic diseases in primary care: protocol of the RedExAP study.

The authors have adequately addressed all the previous comments.

Prior to publication of this manuscript, the authors need to review the year of publication for reference (4); it is stated as BMC Geriatrics 2028.

7. PLOS authors have the option to publish the peer review history of their article (what does this mean?). If published, this will include your full peer review and any attached files.

Reviewer #2: No

---

## [Editor Report · Acceptance letter]

4 Jun 2024

PONE-D-23-38977R2 

PLOS ONE

Dear Dr. Martínez Hortelano, 

I'm pleased to inform you that your manuscript has been deemed suitable for publication in PLOS ONE. Congratulations! Your manuscript is now being handed over to our production team.

Kind regards, 

on behalf of

Dr. Hidetaka Hamasaki 

Academic Editor

PLOS ONE